# IVE-GAN: INVARIANT ENCODING GENERATIVE ADVERSARIAL NETWORKS

## ABSTRACT

Generative adversarial networks (GANs) are a powerful framework for generative tasks. However, they are difficult to train and tend to miss modes of the true data generation process. Although GANs can learn a rich representation of the covered modes of the data in their latent space, the framework misses an inverse mapping from data to this latent space. We propose *Invariant Encoding Generative Adversarial Networks* (IVE-GANs), a novel GAN framework that introduces such a mapping for individual samples from the data by utilizing features in the data which are invariant to certain transformations. Since the model maps individual samples to the latent space, it naturally encourages the generator to cover all modes. We demonstrate the effectiveness of our approach in terms of generative performance and learning rich representations on several datasets including common benchmark image generation tasks.

## 1 INTRODUCTION

Generative Adversarial Networks (GANs) (Goodfellow et al., 2014) emerged as a powerful framework for training generative models in the recent years. GANs consist of two competing (adversarial) networks: a generative model that tries to capture the distribution of a given dataset to map from an arbitrary latent space (usually drawn from a multi-variate Gaussian) to new synthetic data points, and a discriminative model that tries to distinguish between samples from the generator and the true data. Iterative training of both models ideally results in a discriminator capturing features from the true data that the generator does not synthesize, while the generator learns to include these features in the generative process, until real and synthesized data are no longer distinguishable.

Experiments by Radford et al. (2016) showed that a GAN can learn rich representation of the data in the latent space in which interpolations produce semantic variations and shifts in certain directions correspond to variations of specific features of the generated data. However, due to the lack of an inverse mapping from data to the latent space, GANs cannot be used to encode individual data points in the latent space (Donahue et al., 2016).

Moreover, although GANs show promising results in various tasks, such as the generation of realistic looking images (Radford et al., 2015; Berthelot et al., 2017; Sønderby et al., 2016) or 3D objects (Wu et al., 2016), training a GAN in the aforementioned ideal way is difficult to set up and sensitive to hyper-parameter selection (Salimans et al., 2016). Additionally, GANs tend to restrict themselves on generating only a few major modes of the true data distribution, since such a so-called *mode collapse* is not penalized in the GAN objective, while resulting in more realistic samples from these modes (Che et al., 2016). Hence, the majority of the latent space only maps to a few regions in the target space resulting in poor representation of the true data.

We propose a novel GAN framework, *Invariant-Encoding Generative Adversarial Networks* (IVE-GAN), which extends the classical GAN architecture by an additional encoding unit $\mathbf{E}$ to map samples from the true data $\mathbf{x}$ to the latent space $\mathbf{z}$ (compare Fig. 1). To encourage the encoder to learn a rich representation of the data in the latent space, the discriminator $\mathbf{D}$ is asked to distinguish between different predefined transformations $\mathbf{T}(\mathbf{x})$ of the input sample and generated samples $\mathbf{G}(\mathbf{E}(\mathbf{x}))$ by taking the the original input as condition into account. While the discriminator has to learn what the different variations have in common with the original input, the encoder is forced to encode the necessary information in the latent space so that the generator can fool the discriminator by generating samples which are similar to the original samples. Since the discriminator is invariant to the predefined transformations, the encoder can ignore these variations in the input space and learn a

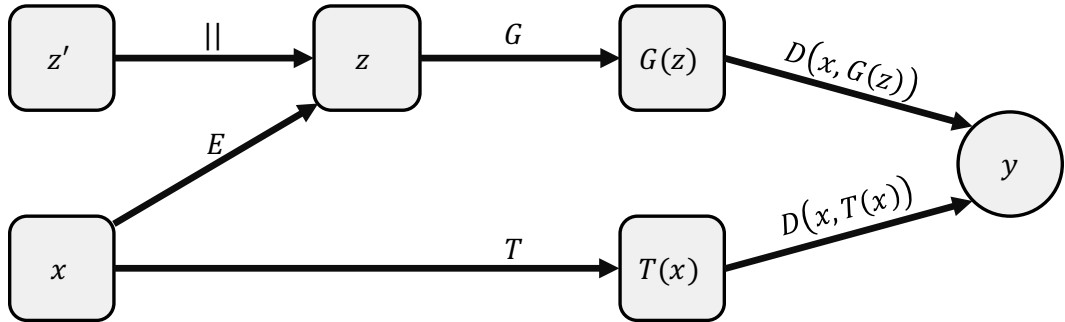

Figure 1: Illustration of the IVE-GAN architecture.

rich and to such transformations invariant representation of the data. The variations of the generated samples are modeled by an additional latent vector $\mathbf{z}'$ (drawn from a multi-variate Gaussian). Thus, the encoded samples condition the generator $G(\mathbf{z}', E(\mathbf{x}))$. Moreover, since the discriminator learns to distinguish between generated samples and variations of the original for each individual sample $\mathbf{x}$ in the data, the latent space can not collapse to a few modes of the true data distribution since it will be easy for the discriminator to distinguish generated samples from original ones if the mode is not covered by the generator. Thus, the proposed IVE-GAN learns a rich and to certain transformations invariant representation of a dataset and naturally encourages the generator to cover all modes of the data distribution.

To generate novel samples, the generator $G(\mathbf{z})$ can be fed with an arbitrary latent representation $\mathbf{z} \sim P_{\text{noise}}$.

In summary, we make the following contributions:

- We derive a novel GAN framework for learning rich and transformation invariant representation of the data in the latent space.

- We show that our GANs reproduce samples from a data distribution without mode collapsing issues.

- We demonstrate robust GAN training across various data sets and showcase that our GAN produces very realistic samples.

## 2 RELATED WORK

Generative Adversarial Networks (Goodfellow et al., 2014) (GANs) are a framework for training generative models. It is based on a *min-max-game* of two competing (adversarial) networks. The generator $G$ tries to map an arbitrary latent space $\mathbf{z} \sim P_Z(\mathbf{z})$ (usually drawn from a multi-variate Gaussian) to new synthetic data points by training a generator distribution $P_G(\mathbf{x})$ that matches the true data distribution $P_{\text{data}}(\mathbf{x})$. The training is performed by letting the generator compete against the second network, the discriminator $D$. The discriminator aims at distinguishing between samples from the generator distribution $P_G(\mathbf{x})$ and real data points from $P_{\text{data}}(\mathbf{x})$ by assigning a probability $y = D(\mathbf{x}) \in [0,1]$. The formal definition of the objective of this min-max-game is given by:

$$\min_G \max_D V(D,G) = \mathbb{E}_{\mathbf{x} \sim P_{\text{data}}} \left[ \log(D(\mathbf{x})] + \mathbb{E}_{\mathbf{z} \sim P_Z} \left[ \log\left(1 - D\left(G\left(\mathbf{z}\right)\right)\right)\right] \tag{1}$$

However, training a GAN on this objective usually results in a generator distribution $P_G(\mathbf{x})$ where large volumes of probability mass collapse onto a few major modes of the true data generation distribution $P_{\text{data}}(\mathbf{x})$ (Che et al., 2016). This issue, often called mode collapsing, has been subject of several recent publications proposing new adjusted objectives to reward a model for higher variety in data generation.

A straightforward approach to control the generated modes of a GAN is to condition it with additional information. Conditional Generative Adversarial Nets (Mirza & Osindero, 2014) utilize additional information such as class-labels to direct the data generation process. The conditioning is

done by additionally feeding the information $y$ into both generator and discriminator. The objective function Eq. (1) becomes:

$$\min_G \max_D V(D, G) = \mathbb{E}_{\mathbf{x} \sim P_{\text{data}}} \left[\log(D\left(\mathbf{x}, \mathbf{y}\right)\right] + \mathbb{E}_{\mathbf{z} \sim P_Z} \left[\log\left(1 - D\left(G\left(\mathbf{z}, \mathbf{y}\right), \mathbf{y}\right)\right)\right] \quad (2)$$

Obviously, such a conditioning is only possible if additional information $y$ is provided.

Che et al. (2017) proposed a framework which adds two regularizers to the classical GAN. The *metric regularizer* trains in addition to the generator $G(z) : Z \to X$ an encoder $E(\mathbf{x}) : X \to Z$ and includes the objective $\mathbb{E}_{\mathbf{x} \sim P_{\text{data}}} \left[d(\mathbf{x}, G \circ E(\mathbf{x}))\right]$ in the training. This additional objective forces the generated modes closer to modes of the true data. As distance measure $d$ they proposed e.g. the pixel-wise $L^2$ distance or the distance of learned features by the discriminator (Dumoulin et al., 2016). To encourage the generator to also target minor modes in the proximity of major modes the objective is extended by a *mode regulizer* $\mathbb{E}_{\mathbf{x} \sim P_{\text{data}}} \left[\log D(G \circ E(\mathbf{x}))\right]$.

Another proposed approach addressing the mode collapsing issue are *unrolled GANs* (Metz et al., 2016). In practice GANs are trained by simultaneously updating $V(D, G)$ in Eq. (1), since explicitly updating G for the optimal D for every step is computational infeasible. This leads to an update for the generator which basically ignores the max-operation for the calculation of the gradients and ultimately encourages mode collapsing. The idea behind unrolled GANs is to update the generator by backpropagating through the gradient updates of the discriminator for a fixed number of steps. This leads to a better approximation of Eq. (1) and reduces mode collapsing.

Another way to avoid mode collapse is the Coulomb GAN, which models the GAN learning problem as a potential field with a unique, globally optimal nash equlibrium (Unterthiner et al., 2017).

Work has also been done aiming at introducing an inverse mapping from data $\mathbf{x}$ to latent space $\mathbf{z}$. Bidirectional Generative Adversarial Networks (BiGANs) (Donahue et al., 2016) and Adversarially Learning Inference (Dumoulin et al., 2016) are two frameworks based on the same idea of extending the GAN framework by an additional encoder $E$ and to train the discriminator on distinguishing joint samples $(\mathbf{x}, E(\mathbf{x}))$ from $(G(\mathbf{z}), \mathbf{z})$ in the data space ($\mathbf{x}$ versus $G(\mathbf{z})$) as well as in the latent space ($E(\mathbf{x})$ versus $\mathbf{z}$). Also the inverse mapping ($E(G(\mathbf{z}))$ is never explicitly computed, the authors proved that in an ideal case the encoder learns to invert the generator almost everywhere ($E = G^{-1}$) to fool the discriminator.

However, upon visual inspection of their reported results (Dumoulin et al., 2016), it appears that the similarity of the original $\mathbf{x}$ to the reconstructions $G(E(\mathbf{x})$ is rather vague, especially in the case of relative complex data such as CelebA (compare appendix A). It seems that the encoder concentrates mostly on prominent features such as gender, age, hair color, but misses the more subtle traits of the face.

## 3 PROPOSED METHOD

Consider a subset $S$ of the domain $D$ that is setwise invariant under a transformation $T : D \to D$ so that $\mathbf{x} \in S \Rightarrow T(\mathbf{x}) \in S$. We can utilize different elements $x \in S$ to train a discriminator on learning the underlying concept of $S$ by discriminating samples $x \in S$ and samples $x \notin S$. In an adversarial procedure we can then train a generator on producing samples $x \in S$.

$S$ could be e.g. a set of higher-level features that are invariant under certain transformation. An example for a dataset with such features are natural images of faces. High-level features like facial parts that are critical to classify and distinguish between different faces are invariant e.g. to small local shifts or rotations of the image.

We propose an approach to learn a mapping from the data to the latent space by utilizing such invariant features $S$ of the data. In contrast to the previous described related methods, we learn the mapping from the data to the latent space not by discriminating the representations $E(\mathbf{x})$ and $\mathbf{z}$ but by discriminating generated samples conditioned on encoded original samples $G(\mathbf{z}', E(\mathbf{x})$ and transformations of an original sample $T(\mathbf{x})$ by taking the original sample as additional information into account. In order to fool the discriminator, the encoder has to extract enough features from the original sample so that the generator can reconstruct samples which are similar to the original one, apart from variations $T$. The discriminator, on the other hand, has to learn which features samples $T(\mathbf{x})$ have in common with the original sample $\mathbf{x}$ to discriminate variations from the original samples and generated samples. To fool a *perfect* discriminator the encoder has to extract all individual features from the original sample so that the generator can produce *perfect* variants of the original.

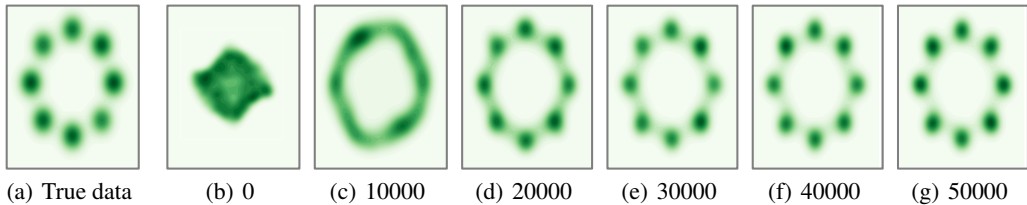

(a) True data    (b) 0    (c) 10000    (d) 20000    (e) 30000    (f) 40000    (g) 50000

Figure 2: Density plots of eight mixtures of Gaussians arranged in a ring. Panel (a) shows the true data and panels (b-g) show the generator distribution at different iteration steps of training.

To generate novel samples, we can draw samples $\mathbf{z} \sim P_Z$ as latent space. To learn a *smooth* representation of the data, we also include such generated samples and train an additional discriminator $D'$ on discriminating them from true data as in the classical GAN objective Eq. (1).

Hence, the objective for the IVE-GAN is defined as a min-max-game:

$$
\begin{aligned}
\min_{G,E} \max_{D,D'} V(D, D', G, E) = \mathbb{E}_{\mathbf{x} \sim P_{\text{data}}} \quad & [\log(D(T(\mathbf{x}), \mathbf{x}) + \log(D'(\mathbf{x})) + \\
\mathbb{E}_{\mathbf{z}' \sim P_{Z'}} \quad & [\log(1 - D(G(\mathbf{z}', E(\mathbf{x})), \mathbf{x}))]] + \\
\mathbb{E}_{\mathbf{z} \sim P_Z, \mathbf{z}' \sim P_{Z'}} & [\log(1 - D'(G(\mathbf{z}', E(\mathbf{x}))))]
\end{aligned}
\tag{3}
$$

One thing to consider here is that by introducing an invariance in the discriminator with respect to transformations $T$, the generator is no longer trying to exactly match the true data generating distribution but rather the distribution of the transformed true data. Thus, one has to carefully decide which transformation $T$ to chose, ideally only such describing already present variations in the data and which are not affecting features in data that are of interest for the representation.

## 4 EXPERIMENTS AND RESULTS

To evaluate the IVE-GAN with respect to quality of the generated samples and learned representations we perform experiments on 3 datasets: a synthetic dataset, the MNIST dataset and the CelebA dataset.

### 4.1 SYNTHETIC DATASET

To evaluate how well a generative model can reproduce samples from a data distribution without missing modes, a synthetic dataset of known distribution is a good way to check if a the model suffers from mode collapsing (Metz et al., 2016).
Following Metz et al. (2016), we evaluate our model on a synthetic dataset of a 2D mixture of eight Gaussians $\mathbf{x} \sim \mathcal{N}(\mu, \boldsymbol{\Sigma})$ with covariance matrix $\boldsymbol{\Sigma}$, and means $\mu_k$ arranged on a ring. As invariant transformation $T(\mathbf{x})$ we define small shifts in both dimensions:

$$
T(\mathbf{x}) := \mathbf{x} + \mathbf{t} \,, \mathbf{t} \sim \mathcal{N}(\mathbf{0}, \, (\boldsymbol{\Sigma}/2)),
\tag{4}
$$

so that the discriminator becomes invariant to the exact position within the eight Gaussians.
Fig. 2 shows the distribution of the generated samples of the model over time compared to the true data generating distribution. The IVE-GAN learns a generator distribution which converges to all modes of the true data generating distribution while distributing its probability mass equally over all modes.

### 4.2 MNIST

As a next step to increase complexity we evaluate our model on the MNIST dataset. As invariant transformations $T(\mathbf{x})$ we define small random shifts (up to 4 pixels) in both width- and height-dimension and small random rotations up to $20°$.
Fig. 3 shows novel generated samples from the IVE-GAN trained on the MNIST dataset as a result

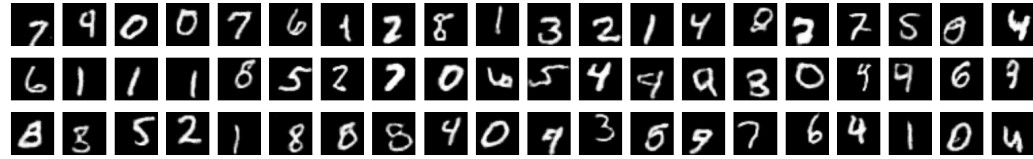

Figure 3: Random samples generated by an IVE-GAN trained on the MNIST dataset. The latent representation (16 dimensions) was randomly drawn from a uniform distribution $\mathbf{z} \sim P_{\text{noise}}$.

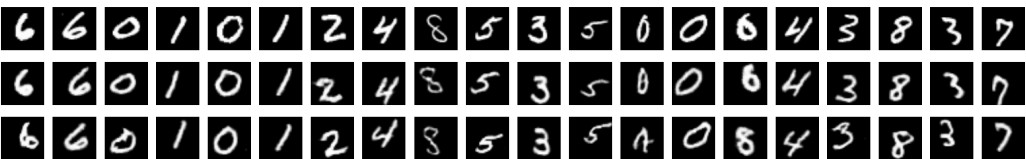

Figure 4: Three rows of MNIST samples. First row: original samples $\mathbf{x}$ from the MNIST training dataset. Second row: generated reconstructions $G(E(\mathbf{x}))$ of the original samples $\mathbf{x}$ from a IVE-GAN with 16-dimensional latent space. Third row: generated reconstructions $G(E(\mathbf{x}))$ of the original samples $\mathbf{x}$ from a IVE-GAN with 3-dimensional latent space.

of randomly sampling the latent representation from a uniform distribution $\mathbf{z} \sim P_{\text{noise}}$. Fig. 4 shows for different samples $\mathbf{x}$ from the MNIST dataset the generated reconstructions $G(\mathbf{z}', E(\mathbf{x}))$ for a model with a 16-dimensional as well as a 3-dimensional latent space $\mathbf{z}$. As one might expect, the model with higher capacity produces images of more similar style to the originals and makes less errors in reproducing digits of unusual style. However, 3 dimensions still provide enough capacity for the IVE-GAN to store enough information of the original image to reproduce the right digit class in most cases. Thus, the IVE-GAN is able to learn a rich representation of the MNIST dataset in 3 dimensions by utilizing class-invariant transformations. Fig. 5 shows the learned representation of the MNIST dataset in 3 dimensions without using further dimensionality reduction methods. We observe distinct clusters for the different digit classes.

### 4.3   CELEBA

As a last experiment we evaluate the proposed method on the more complex CelebA dataset (Liu et al., 2015), centrally cropped to $128 \times 128$ pixel. As in the case of MNIST, we define invariant transformation $T(\mathbf{x})$ as small random shifts (here up to 20 pixel) in both width- and height-dimension as well as random rotations up to $20°$. Additionally, $T(\mathbf{x})$ performs random horizontal flips and small random variations in brightness, contrast and hue.

Fig. 6 shows for different images $\mathbf{x}$ from the CelebA dataset some of the random transformation $T(\mathbf{x})$ and some of the generated reconstructed images $G(\mathbf{z}', E(\mathbf{x}))$ with random noise $\mathbf{z}'$. The reconstructed images show clear similarity to the original images not only in prominent features but also subtle facial traits.

Fig. 7 shows novel generated samples from the IVE-GAN trained on the CelebA dataset as an result of randomly sampling the latent representation from a uniform distribution $\mathbf{z} \sim P_{\text{noise}}$. To illustrate the influence of the noise component $\mathbf{z}'$, the generation was performed with the same five noise components for each image respectively. We observe that altering the noise component induces a similar relative transformation in each image.

To visualize the learned representation of the trained IVE-GAN we encode 10.000 samples from the CelebA dataset into the 1024-dimensional latent space and projected it into two dimensions using *t-Distributed Stochastic Neighbor Embedding* (t-SNE) (Maaten & Hinton, 2008).

Fig. 8(a) shows this projection of the latent space with example images for some high density regions. Since the CelebA dataset comes with labels, we can evaluate the representation with respect to its ability to clusters images of same features. Fig. 8(b)-Fig. 8(e) shows the t-SNE embedding of both the latent representation and the original images for a selection of features. Observing the

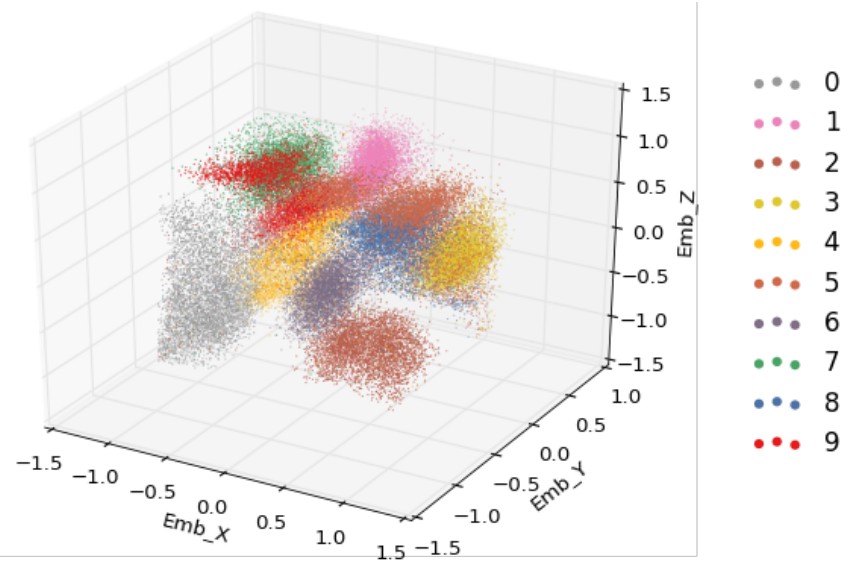

Figure 5: Representation of the MNIST dataset using the 3-dimensional latent space learned by the IVE-GAN. The colors correspond to the digit labels.

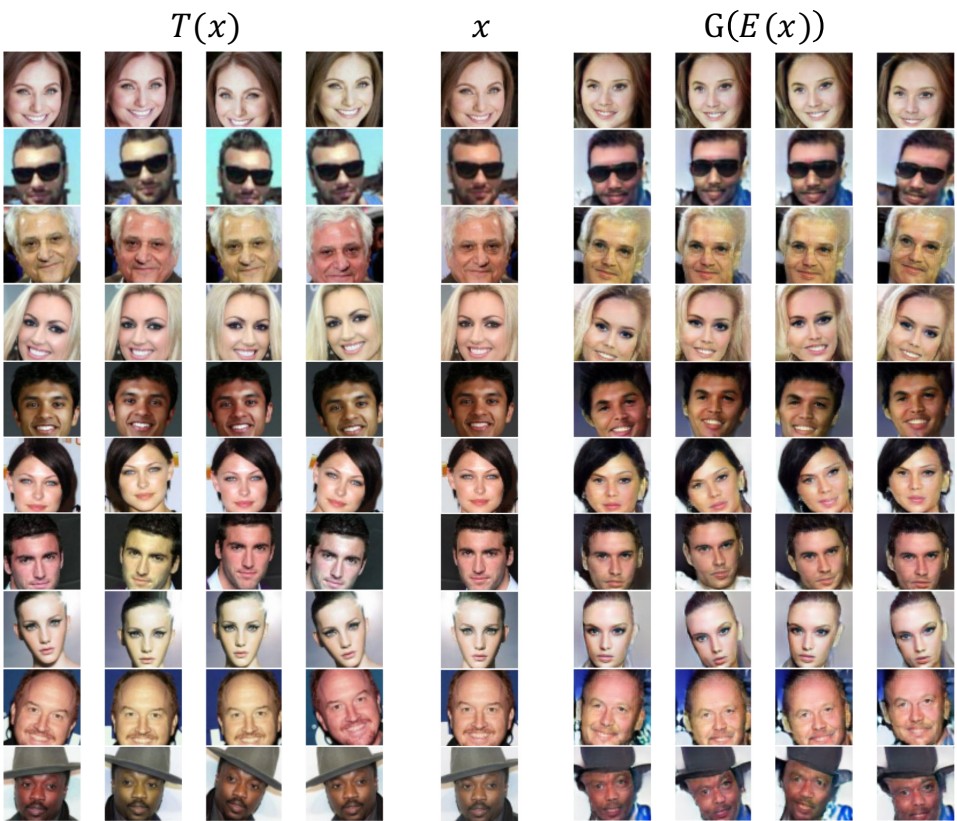

Figure 6: Original samples $\mathbf{x}$ from the CelebA dataset, their random transformation $T(\mathbf{x})$ and the generated reconstructions $G(E(\mathbf{x}))$.

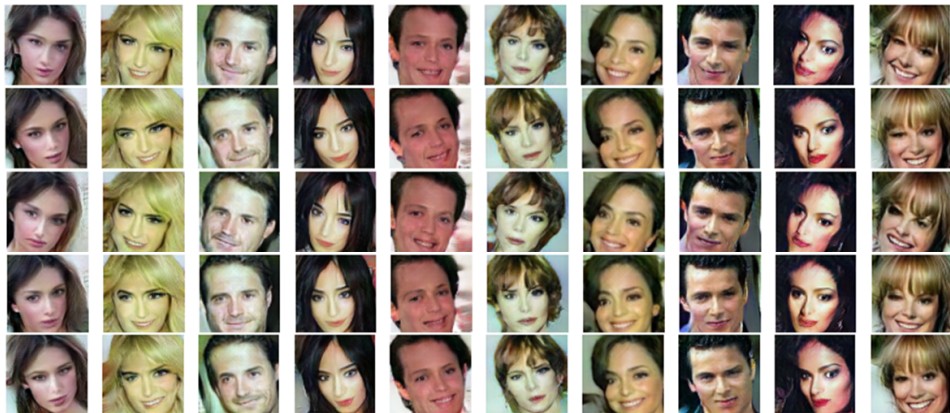

Figure 7: Generated samples with same randomly drawn latent representation $\mathbf{z} \sim P_{\text{noise}}$ vertically and same randomly drawn noise component $\mathbf{z}' \sim P_{\text{noise}}$ horizontally.

visualization of the latent space, we can make out distinct clusters of images sharing similar style and features. Images that are close together in the latent space, share similar visual attributes. It is noteworthy that even images of people wearing normal eyeglasses are separated from images of people wearing sunglasses. By comparing the embedding of the learned representation with the embedding of the original images we observe a clear advantage of the representation learned by the IVE-GAN in terms of clustering images with the same features.

We also evaluate whether smooth interpolation in the learned feature space can be performed. Since we have a method at hand that can map arbitrary images from the dataset into the latent space, we can also interpolate between arbitrary images of this dataset. This is in contrast to Radford et al.(2016), who could only show such interpolations between generated ones.

Fig. 9 shows generated images based on interpolation in the latent representation between two original images. The intermediate images, generated from the interpolated latent representations are visually appealing and show a smooth transformation between the images. This finding indicates that the IVE-GAN learns a smooth latent space and can generate new realistic images not only from latent representations of training samples.

## 5 CONCLUSION

With this work we proposed a novel GAN framework that includes a encoding unit that maps data to a latent representation by utilizing features in the data which are invariant to certain transformations. We evaluate the proposed model on three different dataset and show the IVE-GAN can generate visually appealing images of high variance while learning a rich representation of the dataset also covering subtle features.

ACKNOWLEDGMENTS

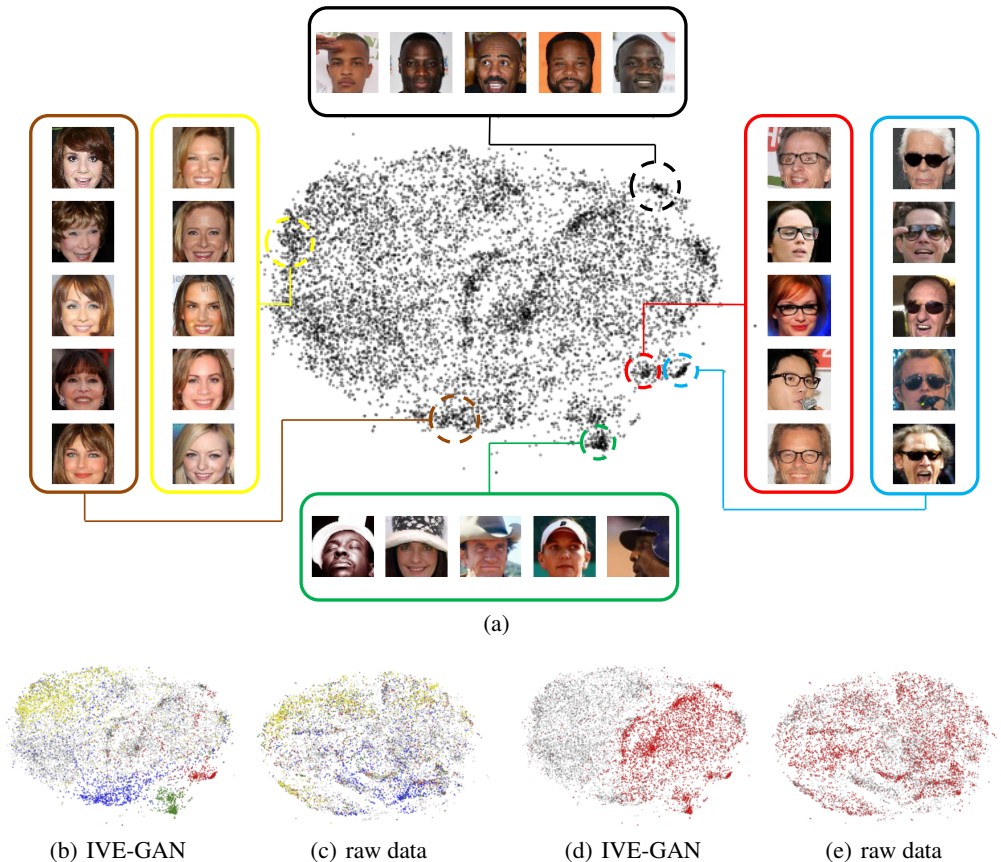

(a)

(b) IVE-GAN  (c) raw data  (d) IVE-GAN  (e) raw data

Figure 8: Visualization of the 2-dimensional t-SNE of the 1024-dimensional latent representation of 10.000 CelebA images. (**a**): Example images for some of the high-density regions of the embedding. (**b**): t-SNE embedding of the latent representation colored accordingly to some labels from the CelebA dataset. red: eyeglasses, green: wearing hat, blue: bangs, yellow: blond hair. (**c**): t-SNE embedding of the original CelebA raw data, same color code as in panel (b). (**d**): t-SNE embedding of the latent representation colored accordingly to the gender. Red: male. (**e**): t-SNE embedding of the original CelebA raw data, same color code as in panel (d).

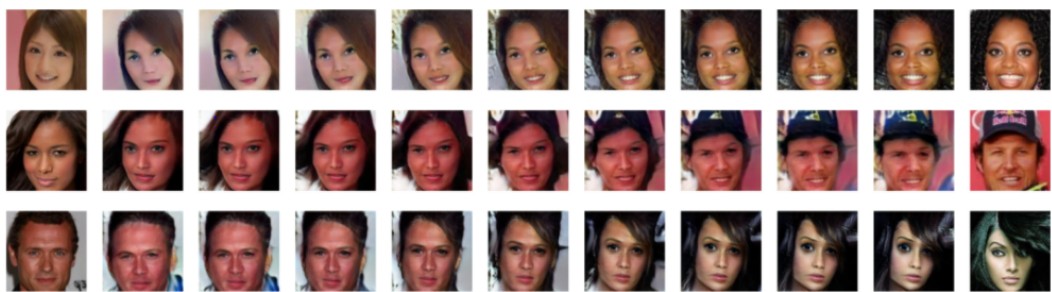

Figure 9: Illustration of interpolation in the latent space between 3 pairs of original images respectively. The first and the last image in each row are original images from the CelebA dataset. The intermediate images are generated reconstructions based on step-wise interpolating between the latent representation of the respective original images.

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

## A  GENERATED IMAGES BY ADVERSARIALLY LEARNED INFERENCE (ALI)

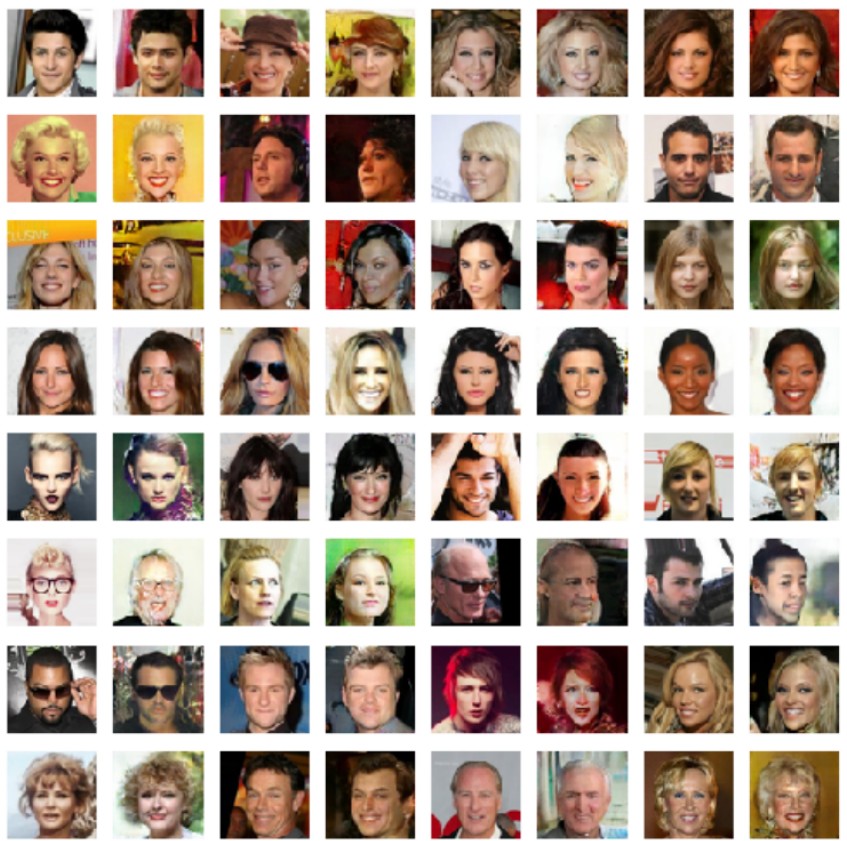

Figure 10: Samples and reconstructions on the CelebA dataset by ALI. Odd columns are original samples and even columns are corresponding reconstructions. Images taken from the original paper (Dumoulin et al., 2016).

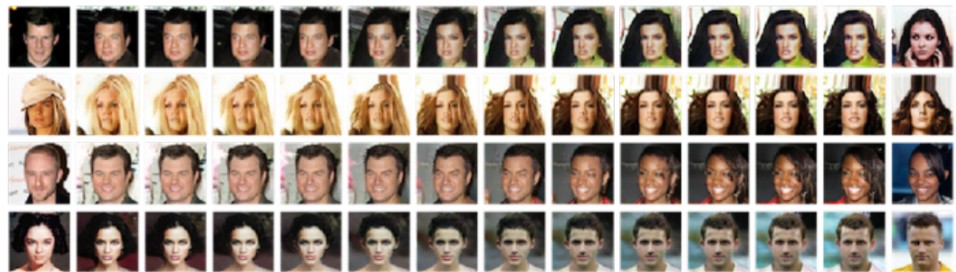

Figure 11: Latent space interpolations on the CelebA dataset by ALI. Images taken from the original paper (Dumoulin et al., 2016).

# B   NETWORK ARCHITECTURE AND HYPERPARAMETERS

Table 1: Network architecture and hyperparameters for the synthetic dataset experiment.

| Unit | Operation | Num Neurons | Activation |
|------|-----------|-------------|------------|
| $E(\mathbf{x})$ | | | |
| | Dense | 128 | Tanh |
| | Dense | 2 | Tanh |
| $G(\mathbf{z}'E(\mathbf{x}))$ | | | |
| | Concatenate $\mathbf{z}'$ and $z = E(\mathbf{x})$ | | |
| | Dense | 128 | Tanh |
| | Dense | 2 | Tanh |
| $D(x, \cdot)$ | | | |
| | Concatenate $\mathbf{x}$ and $T(\mathbf{x})$ respectively $G(E(\mathbf{x}))$ | | |
| | Dense | 128 | Tanh |
| | Dense | 1 | Tanh |
| $D'(D_{\mathrm{enc}}(\cdot))$ | | | |
| | Dense | 128 | Tanh |
| | Dense | 1 | Tanh |

| | |
|---|---|
| $z'$-dimensions | 4 |
| $z$-dimensions | 3 |
| Optimizer$_G$ | Adam ($\alpha = 2 \cdot 10^{-4}$, $\beta_1 = 0.7$, $\beta_2 = 0.999$, $\epsilon = 10^{-8}$) |
| Optimizer$_D$ | Adam ($\alpha = 1 \cdot 10^{-4}$, $\beta_1 = 0.7$, $\beta_2 = 0.999$, $\epsilon = 10^{-8}$) |
| Batch size | 1024 |
| Epochs | 50000 |
| LReLU slope | 0.2 |
| Weight initialization | LReLU-layer: He, else: Xavier |
| Bias initialization | Constant zero |

Table 2: Network architecture and hyperparameters for the MNIST experiment.

| Unit | Operation | Kernel | Strides | Num Filter | BN? | Activation |
|---|---|---|---|---|---|---|
| $E(\mathbf{x})$ | | | | | | |
| | Conv | 5x5 | 2x2 | 64 | ✓ | LReLU |
| | Conv | 3x3 | 2x2 | 128 | ✓ | LReLU |
| | Conv | 3x3 | 2x2 | 256 | ✓ | LReLU |
| | Conv | 3x3 | 2x2 | 256 | ✓ | LReLU |
| | Conv | 2x2 | 1x1 | 256 | ✓ | LReLU |
| | Flatten | | | | | |
| | Dense | | | 1024 | ✗ | Tanh |
| $G(\mathbf{z}'E(\mathbf{x}))$ | | | | | | |
| | Concatenate $\mathbf{z}'$ and $z = E(\mathbf{x})$ | | | | | |
| | Dense | | | 4096 | ✓ | LReLU |
| | Reshape to [Batch Size, 4, 4, 256] | | | | | |
| | Conv transposed | 3x3 | 2x2 | 256 | ✓ | LReLU |
| | Conv transposed | 3x3 | 2x2 | 128 | ✓ | LReLU |
| | Conv transposed | 5x5 | 2x2 | 64 | ✗ | LReLU |
| | Conv transposed | 5x5 | 1x1 | 1 | ✗ | Sigmoid |
| $D_{\mathrm{enc}}(\cdot)$ | | | | | | |
| | Conv | 5x5 | 2x2 | 64 | ✓ | LReLU |
| | Conv | 3x3 | 2x2 | 128 | ✓ | LReLU |
| | Conv | 3x3 | 2x2 | 256 | ✓ | LReLU |
| | Conv | 3x3 | 2x2 | 256 | ✓ | LReLU |
| | Conv | 2x2 | 1x1 | 256 | ✓ | LReLU |
| | Flatten | | | | | |
| | Dense | | | 3 | ✗ | LReLU |
| $D(x, D_{\mathrm{enc}}(\cdot))$ | | | | | | |
| | Concatenate $D_{\mathrm{enc}}(\mathbf{x})$ and $D_{\mathrm{enc}}(\cdot)$ | | | | | |
| | Dense | | | 128 | ✗ | LReLU |
| | Dense | | | 64 | ✗ | LReLU |
| | Dense | | | 16 | ✗ | LReLU |
| | Dense | | | 1 | ✗ | Linear |
| $D'(D_{\mathrm{enc}}(\cdot))$ | | | | | | |
| | Dense | | | 64 | ✗ | LReLU |
| | Dense | | | 1 | ✗ | Linear |

| | |
|---|---|
| $z'$-dimensions | 4 |
| $z$-dimensions | 3 |
| Optimizer$_G$ | Adam ($\alpha = 2 \cdot 10^{-4}$, $\beta_1 = 0.7$, $\beta_2 = 0.999$, $\epsilon = 10^{-8}$) |
| Optimizer$_D$ | Adam ($\alpha = 1 \cdot 10^{-4}$, $\beta_1 = 0.7$, $\beta_2 = 0.999$, $\epsilon = 10^{-8}$) |
| Batch size | 512 |
| Epochs | 100 |
| LReLU slope | 0.2 |
| Weight initialization | LReLU-layer: He, else: Xavier |
| Bias initialization | Constant zero |

Table 3: Network architecture and hyperparameters for the CelebA experiment.

| Unit | Operation | Kernel | Strides | Num Filter | BN? | Activation |
|---|---|---|---|---|---|---|
| $E(\mathbf{x})$ | | | | | | |
| | Conv | 5x5 | 2x2 | 128 | ✓ | LReLU |
| | Conv | 5x5 | 2x2 | 128 | ✓ | LReLU |
| | Conv | 5x5 | 2x2 | 256 | ✓ | LReLU |
| | Conv | 3x3 | 2x2 | 256 | ✓ | LReLU |
| | Conv | 3x3 | 2x2 | 512 | ✓ | LReLU |
| | Conv | 3x3 | 2x2 | 512 | ✓ | LReLU |
| | Conv | 2x2 | 1x1 | 1024 | ✓ | LReLU |
| | Flatten | | | | | |
| | Dense | | | 1024 | ✗ | Tanh |
| $G(\mathbf{z}'E(\mathbf{x}))$ | | | | | | |
| | Concatenate $\mathbf{z}'$ and $z = E(\mathbf{x})$ | | | | | |
| | Dense | | | 4096 | ✓ | LReLU |
| | Reshape to [Batch Size, 2, 2, 1024] | | | | | |
| | Conv transposed | 2x2 | 2x2 | 512 | ✓ | LReLU |
| | Conv transposed | 2x2 | 1x1 | 512 | ✓ | LReLU |
| | Conv transposed | 3x3 | 2x2 | 256 | ✓ | LReLU |
| | Conv transposed | 3x3 | 1x1 | 256 | ✓ | LReLU |
| | Conv transposed | 3x3 | 2x2 | 256 | ✓ | LReLU |
| | Conv transposed | 3x3 | 1x1 | 256 | ✓ | LReLU |
| | Conv transposed | 5x5 | 2x2 | 128 | ✓ | LReLU |
| | Conv transposed | 5x5 | 1x1 | 128 | ✓ | LReLU |
| | Conv transposed | 5x5 | 2x2 | 128 | ✓ | LReLU |
| | Conv transposed | 5x5 | 1x1 | 128 | ✗ | LReLU |
| | Conv transposed | 5x5 | 2x2 | 64 | ✗ | LReLU |
| | Conv transposed | 5x5 | 1x1 | 3 | ✗ | Sigmoid |
| $D_{\mathrm{enc}}(\cdot)$ | | | | | | |
| | Conv | 5x5 | 2x2 | 128 | ✓ | LReLU |
| | Conv | 5x5 | 2x2 | 128 | ✓ | LReLU |
| | Conv | 5x5 | 2x2 | 256 | ✓ | LReLU |
| | Conv | 3x3 | 2x2 | 256 | ✓ | LReLU |
| | Conv | 3x3 | 2x2 | 512 | ✓ | LReLU |
| | Conv | 3x3 | 2x2 | 512 | ✓ | LReLU |
| | Conv | 2x2 | 1x1 | 1024 | ✓ | LReLU |
| | Flatten | | | | | |
| | Dense | | | 1024 | ✗ | LReLU |
| $D(x, D_{\mathrm{enc}}(\cdot))$ | | | | | | |
| | Concatenate $D_{\mathrm{enc}}(\mathbf{x})$ and $D_{\mathrm{enc}}(\cdot)$ | | | | | |
| | Dense | | | 1024 | ✗ | LReLU |
| | Dense | | | 512 | ✗ | LReLU |
| | Dense | | | 128 | ✗ | LReLU |
| | Dense | | | 1 | ✗ | Linear |
| $D'(D_{\mathrm{enc}}(\cdot))$ | | | | | | |
| | Dense | | | 128 | ✗ | LReLU |
| | Dense | | | 1 | ✗ | Linear |

| | |
|---|---|
| $z'$-dimensions | 16 |
| $z$-dimensions | 1024 |
| Optimizer$_G$ | Adam ($\alpha = 2 \cdot 10^{-4}$, $\beta_1 = 0.5$, $\beta_2 = 0.999$, $\epsilon = 10^{-8}$) |
| Optimizer$_D$ | Adam ($\alpha = 1 \cdot 10^{-4}$, $\beta_1 = 0.5$, $\beta_2 = 0.999$, $\epsilon = 10^{-8}$) |
| Batch size | 64 |
| Epochs | 16 |
| LReLU slope | 0.2 |
| Weight initialization | LReLU-layer: He, else: Xavier |
| Bias initialization | Constant zero |

