# OpenReview forum: "IVE-GAN: Invariant Encoding Generative Adversarial Networks"
_ICLR.cc/2018/Conference — Reject_

### Official Review · AnonReviewer2 · 2017-11-27
**A novel algorithm, but not enough justification**

**Rating:** 5
**Confidence:** 4

**Review:**

The paper proposes a modified GAN objective, summarized in Eq.(3). It consists of two parts:
(A) Classic GAN term: \E_{x ~ P_{data} } \log D'(x) + \E_{z ~ P_{Z}, z' ~ P_{Z'}  } \log D'( G(z',E(x))   )
(B) Invariant Encoding term: \E_{x ~ P_{data} }  [ \log D(T(x),x) + \E_{z' ~ P_{Z'}  } \log D( G(z',E(x)), x   ) ]

Term (A) is standard, except the latent space of original GAN is decomposed into (1) the feature, which should be invariant between x and T(x), and (2) the noise, which is for the diversity of generated x.

Term (B) is the proposed invariant-encoding scheme. It is essentially a conditional GAN, where the the generated sample G(z',E(x)) is conditioned on input sample x, which guarantees that the generated sample is T(x) of x.
In fact, this could be theoretically justified. Suggestion: the authors might want to follow the proofs of Proposition 1 or 2 in [*] to show similar conclusion, making the paper stronger.

[*] ALICE: Towards Understanding Adversarial Learning for Joint Distribution Matching, NIPS 2017

The definition of feature-invariant is E(T(x))=E(x)=E(G(z',E(x))), while the objective of the paper achieves T(x)=G(z',E(x)). Applying E() to both side yields the invariant features.  It might be better to make this point clear.

Overall Comments:

Originality: the proposed IVE-GAN algorithm is quite novel.
Quality: The paper could be stronger, if the theoretical justification has been provided.
Clarity: Overall clear, while important details are missing. Please see some points in Detailed Comments.
Significance: The idea is interesting, it would be better if the quantitative evidence has been provided to demonstrate the use of the learned invariant feature. For example, some classification task to demonstrate the learned rotation-invariant feature shows higher accuracy.

Detailed Comments:

-- In Figure 1, please explains that "||" is the concatenation operator for better illustration.

-- When summarizing the contributions of the paper, it is mentioned that "our GANs ... without mode collapsing issues". This is a strong point to claim. While precisely identifying the "mode collapsing issue" itself is difficult, the authors only show that samples in all modes are generated on the toy datasets. Please consider to rephrase.

-- In Section 2, y is first used to indicate true/false of x in Eq.(1), then y is used to indicate the associated information (e.g., class label) of x in Eq.(2). Please consider to avoid overloading notations.

-- In Eq.(3), the expectation \E_{z ~ P_Z} in the 3rd term is NOT clear, as z is not involved in the evaluation. I guess it may be implemented as z=E(x), where x ~ P_{data}. From the supplement tables, It seems that the novel sample G(z', E(x)) is implemented as G evaluated on the concatenation of noise sample z' ~ P_{Z'} and encoded feature z=E(x).
I am wondering how to generate novel samples? Related to this,  Please clarify how to implement: "To generate novel samples, we can draw samples z ~ P_Z as latent space".

-- Section 5, "include a encoding unit" ---> "an"

-- In Supplement, please revise G(z'E(x)) to G(z', E(x)) in every table.

---

### Official Review · AnonReviewer1 · 2017-11-28
**weak evaluation**

**Rating:** 4
**Confidence:** 5

**Review:**

This paper presents the IVE-GAN, a model that introduces en encoder to the Generative Adversarial Network (GAN) framework. The model is evaluated qualitatively through samples and reconstructions on a synthetic dataset, MNIST and CelebA.

Summary:
The evaluation is superficial, no quantitative evaluation is presented and key aspects of the model are not explored. Overall, there just is not enough innovation or substance to warrant publication at this point.

Impact:
The motivation given throughout the introduction -- to add an encoder (inference) network to GANs -- is a bit odd in the light of the existing literature. In addition to the BiGAN/ALI models that were cited, there are a number of (not cited) papers with various ways of combining GANs with VAE encoders to accomplish exactly this. If your goal was to improve reconstructions in ALI, one could simply add an reconstruction (or cycle) penalty to the ALI objective as advocated in the (not cited) ALICE paper (Li et al., 2017 -- "ALICE: Towards Understanding Adversarial Learning for Joint Distribution Matching").

The training architecture presented here is novel as far as I know, though I am unconvinced that it represents an optimum in model design space. The model presented in the ALICE paper would seem to be a more elegant solution to the motivation given in this paper.

Model Feature:
The authors should discuss in detail the interaction between the regular GAN pipeline and the introduced variant (with the transformations). Why is the standard GAN objective thrown in? I assume it is to allow you to sample directly from the noise in z (as opposed to z' which is used for reconstruction), but this is not discussed in much detail. The GAN objective and the added IVE objective seem like they will interact in not all together beneficial ways, with the IVE component pushing to make the distribution in z complicated. This would result in a decrease in sample quality. Does it? Exploration of this aspect of the model should be included in the empirical evaluation.

Also the addition of the transformations added to the proposed IVE pipeline seem to cause the latent variations z' to encode these variations rather than the natural variations that exist in the dataset. They would seem to make it difficult to encode someone face and make some natural manipulates (such as adjusting the smile) that are not included in this transformations.

Empirical Evaluation:
Comparison to BiGAN/ALI: The authors motivate their work by drawing comparisons to BiGAN/ALI, showing CelebA reconstructions from the ALI paper in the appendix. The comparison is not fair for two reasons, (1) authors should state that their reconstructions are made at a higher resolution (seems like 128x128, which is now standard but was not so when the BiGAN/ALI papers came out, they were sampled at 64x64), also, unlike the ALI results, they authors cut the background away from the CelebA faces. This alone could account for the difference between the two models, as ICE-GAN only has to encode the variability extant in faces and hair, ALI had to additionally encode the much greater variability in the background. The failure to control the experimental conditions makes this comparison inappropriate.

There is no quantitative evaluations at all. While many GAN papers do not place an emphasis on quantitative evaluations, at this point, I consider the complete lack of such an evaluation as a weakness of the paper.

Finally, based on just the samples offered in the paper, which is admittedly a fairly weak standard, the model does not seem to be among the state-of-the-art on CelebA that have been reported in the literature. Given the rapid progress that is being made, I do not feel this should be could against this particular paper, but the quality of samples cannot be considered a compelling reason to accept the paper.

Minor comment:
The authors appear to be abusing the ICLR style file by not leaving a blank line  between paragraphs. This is annoying and not at all necessary since ICLR does not have a strict page limit.

Figure 1 is not consistent with the model equations (in Eqns. 3). In particular, Figure 1 is missing the standard GAN component of the model.

I assume that the last term in Eqns 3 should have G(z) as opposed to G(z',E(x)). Is that right?

---

### Official Review · AnonReviewer3 · 2017-11-29
**Learning factored representations with GANs**

**Rating:** 5
**Confidence:** 4

**Review:**

This paper proposes a GAN-based approach to learning factored latent representations of images. The latent space is factored into a set of variables encoding image identity and another set of variables encoding deformations of the image. Variations on this theme have been presented previously (see, e.g. "Learning to Generate Chairs with Convolutional Neural Networks" by Dosvitskiy et al., CVPR 2015). There's also some resemblance to methods for learning unsupervised embeddings based on discrimination between "invariance classes" of images (see, e.g. "Discriminative Unsupervised Feature Learning with Convolutional Neural Networks" by Dosovitskiy et al., NIPS 2014). The submitted paper is the first GAN-based paper on this precise topic I've seen, but it's hard to keep up these days.

The method described in the paper applies existing ideas about learning factored image representations which split features into subsets describing image type and variation. The use of GANs is a somewhat novel extension of existing conditional GAN methods. The paper is reasonably written, though many parentheses are missing. The qualitative results seem alright, and generally comparable to concurrent work on GANs. No concrete tasks or quantitative metrics are provided. Perhaps the authors could design a simple classification-based metric using a k-NN type classifier on top of the learned representations for the MNIST or CelebA tasks. The t-SNE plots suggest a comparison with the raw data will be favourable (though stronger baselines would also be needed).

I'm curious why no regularization (e.g., some sort of GAN cost) was placed on the marginal distribution of the "image type" encodings E(x). It seems like it would be useful to constrain these encodings to be distributionally similar to the prior from which free samples will be drawn.

---

### Public Comment · (anonymous) · 2017-11-18
**Related to Data Augmentation Generative Adversarial Networks?**

Different motiviation, but it seems like https://arxiv.org/abs/1711.04340 follow a similar approach. Can the authors comment on this?

---

> ### Author Response · Authors · 2017-11-20
> **a subcase of our proposed method**
>
> Thanks a lot for bringing this paper to our attention. It is interesting to see that now also another group comes up with a methodology we originally proposed. Indeed, Antoniou et al. basically exploit the same idea as we propose: utilizing different variations of the same entity to learn a representation of the entity itself. Apart from the motivation (they use the method for augmentation, we focus on unsupervised representation learning), the main difference is that we produce the variations by defining transformations under which the true data generating distribution is invariant (such as small rotations or shifts of an image). In contrast, the referenced paper utilizes different samples with the same class membership. The latter procedure is a subcase of a setwise invariant transformation we describe in our paper, where a set is explicitly defined by its class membership and the transformation is indirectly given by the different samples within the set. This is a subcase of our proposed (general) definition and can obviously only be utilized if class membership is given, thus being unsuitable for an unsupervised setting.

---

### Decision · Program_Chairs · 2018-01-29
**ICLR 2018 Conference Acceptance Decision**

**Decision:**

Reject

**Comment:**

Reviewers recognize that the method proposed is somewhat novel but have strong reservations on the experimental evaluation. Discussion of some relevant papers is also missing (eg, Li et al, 2017 : ALICE). The authors have not responded to the many concerns expressed by the reviewers.